behaviour, evolution, ecology

extra-pair paternity, inbreeding, inclusive fitness, kinship, parental care, parent–offspring relatedness

**Author for correspondence:**
Elizabeth A. Gow
e-mail: elizabeth.gow@gmail.com

†Present address: Department of Integrative Biology, University of Guelph, Guelph, ON, Canada N1G 2W1.

# Testing predictions of inclusive fitness theory in inbreeding relatives with biparental care

Elizabeth A. Gow[1,†], Peter Arcese[1], Danielle Dagenais[2], Rebecca J. Sardell[3], Scott Wilson[1,4] and Jane M. Reid[3,5]

[1]Department of Forest and Conservation Sciences, University of British Columbia, Vancouver, British Columbia, Canada
[2]Natural Resources and Environmental Studies, University of Northern British Columbia, Prince George, British Columbia, Canada
[3]School of Biological Sciences, University of Aberdeen, Aberdeen, Scotland
[4]National Wildlife Research Centre, Environment and Climate Change Canada, Ottawa, Ontario, Canada
[5]Centre for Biodiversity Dynamics, Norwegian University of Science and Technology, Trondheim, Norway

(iD) EAG, 0000-0001-8890-4503; JMR, 0000-0002-5007-7343

Inclusive fitness theory predicts that parental care will vary with relatedness between potentially caring parents and offspring, potentially shaping mating system evolution. Systems with extra-pair paternity (EPP), and hence variable parent–brood relatedness, provide valuable opportunities to test this prediction. However, existing theoretical and empirical studies assume that a focal male is either an offspring's father with no inbreeding, or is completely unrelated. We highlight that this simple dichotomy does not hold given reproductive interactions among relatives, complicating the effect of EPP on parent–brood relatedness yet providing new opportunities to test inclusive fitness theory. Accordingly, we tested hierarchical hypotheses relating parental feeding rate to parent–brood relatedness, parent kinship and inbreeding, using song sparrows (*Melospiza melodia*) experiencing natural variation in relatedness. As predicted, male and female feeding rates increased with relatedness to a dependent brood, even controlling for brood size. Male feeding rate tended to decrease as paternity loss increased, and increased with increasing kinship and hence inbreeding between socially paired mates. We thereby demonstrate that variation in a key component of parental care concurs with subtle predictions from inclusive fitness theory. We additionally highlight that such effects can depend on the underlying social mating system, potentially generating status-specific costs of extra-pair reproduction.

## 1. Introduction

A central ambition in evolutionary ecology is to understand how 'altruistic' behaviours, which cost actors but benefit recipients, evolve as functions of interactions among relatives [1–4]. Parental care provided to dependent offspring represents one critically important altruistic behaviour that simultaneously emerges from, and can shape ongoing evolution of, complex reproductive strategies and mating systems. Variable parental care, therefore, provides one long-standing focus for developing and testing inclusive fitness theory [5].

Parental or alloparental care is typically predicted to increase with a focal adult's relatedness to dependent offspring, following the basic principle of Hamilton's rule [1,3,4,6]. Systems where relatedness between potentially caring adults and dependent offspring varies among family groups offer interesting opportunities to test this prediction, and to examine the degree to which adaptive plastic responses in parental care can arise and potentially shape mating system evolution. Such variation in adult–offspring relatedness is commonplace in socially monogamous systems with variable extra-pair paternity (EPP) [7–11]. Here,

potentially caring males might not sire all offspring produced by their socially paired female [12]. All else being equal, paternal care is then predicted to increase with a male's paternity success and resulting male–brood relatedness, defined as the total number of copies of an allele that is present in focal male $i$ that is expected to be present in the brood (hereafter 'total allelic value', $TAV_i$) [5,12–15]. Decreased paternal care following paternity loss can then create a cost of female extra-pair reproduction that could be sufficient to constrain the evolution of underlying polyandry [16]. Systems characterized by social monogamy, biparental care and EPP are consequently interesting systems where evolutionary dynamics of parental care and mating system are directly intertwined, attracting substantial theory development [13,14,17–19] and empirical tests [6,20–24].

Yet existing theoretical and empirical studies typically assume that the relatedness between a potentially caring male and a dependent offspring is either ½ or 0, meaning the male is either the offspring's father with no inbreeding, or is completely unrelated [13,14,19,25,26]. The male's total 'relatedness' to a dependent brood, or $TAV_i$, is then simply ½BS • $P_{WPO}$ where BS is brood size and $P_{WPO}$ is the proportion of the brood that are genetic offspring of the focal male (i.e. within-pair offspring, WPO). This expression reduces to ½$N_{WPO}$, where $N_{WPO}$ is the number of WPO [12]. Similarly, the decrease in $TAV_i$ to a potentially caring male resulting from EPP is simply ½BS • $P_{EPO}$, where $P_{EPO}$ is the proportion of the brood that are extra-pair offspring (EPO, hence $P_{EPO} = 1 - P_{WPO}$). However, these basic premises may not hold in reality, complicating the effect of EPP on parent–brood relatedness and associated optimal allocations of parental care.

Specifically, many populations and mating systems foster reproductive interactions among multiple relatives, including active or passive inbreeding and different forms of kin-structured reproductive groups or neighbourhoods [9–11,27,28]. Such systems can generate more subtle forms of variation in adult–offspring relatedness than a simple 'parent or not' dichotomy. Specifically, a focal male $i$ that fails to sire an offspring of his socially paired female $j$ could still be related to that EPO, and hence accrue some inclusive fitness benefit of paternal care, if he is related to the EPO's mother (i.e. his socially paired female) by coefficient of kinship $k_{ij} > 0$, and/or to the EPO's genetic father (i.e. his socially paired female's extra-pair mate $q$) by $k_{iq} > 0$ [12,29] (electronic supplementary material, appendix S1). Quantitatively, a male's relatedness to an EPO that he did not sire but could rear is $r_{iEPO} = k_{ij} + k_{iq}$ [12]. Furthermore, the general expressions for relatedness between a focal male and female and their WPO are $r_{iWPO} = ½ + k_{ij} + ½f_i$ and $r_{jWPO} = ½ + k_{ij} + ½f_j$, respectively, where $f_i$ and $f_j$ are these parents' own coefficients of inbreeding [12,30] (electronic supplementary material, appendix S1). Similarly, a female's relatedness to its EPO is $r_{jEPO} = ½ + k_{jq} + ½f_j$, where $k_{jq}$ is the coefficient of kinship between $j$ and $q$. These expressions show that a focal parent can be considerably more closely related to its offspring than the basic value of ½ when it is related to its mate ($k_{ij} > 0$ or $k_{jq} > 0$) and/or is inbred itself ($f_i > 0$ or $f_j > 0$), and hence given inbreeding in the current and/or previous generation [12,31–33]. Consequently, the total relatedness between a potentially caring male and a focal dependent brood, $TAV_i$, is most generally calculated as the sum of $r_{iWPO}$ or $r_{iEPO}$ across all WPO and EPO within the brood, respectively, whereas $TAV_j$ for a female is the sum of $r_{jWPO}$ and $r_{jEPO}$ across all these offspring [12] (electronic supplementary

material, appendix S1). TAV for a focal brood can, therefore, differ between paired males and females, and can substantially exceed the typically assumed basic values of ½$N_{WPO}$ and ½BS, respectively [12]. Furthermore, because a potentially caring male's relatedness to an EPO may not be zero, the decrease in $TAV_i$ resulting from EPP no longer simply equals ½BS • $P_{EPO}$. Rather, this difference (hereafter 'lost allelic value', LAV) can be calculated as LAV = PAV – $TAV_i$, where PAV is the 'potential allelic value' of the brood to the male if he had sired the entire brood (electronic supplementary material, appendix S1). Subtle patterns of adaptive variation in the degree of parental care might then be predicted, such that paternal care might increase more tightly with increasing TAV than with BS, and decrease with increasing LAV (and $P_{EPO}$, table 1), reflecting the fundamental premise that care should be adjusted in proportion to relatedness to dependent offspring. Such subtle modulation of parental care might then further affect mating system dynamics emerging among interacting relatives [13,14,18,19].

Additionally, recent advances in inclusive fitness theory predict that the kinship $k_{ij}$ between paired parents will directly influence optimal parental investment [31]. Specifically, if parental care, which forms a component of parental investment, can ameliorate inbreeding depression in offspring viability, then the optimal degree of care is predicted to increase with increasing $k_{ij}$ [31]. By contrast, under these circumstances, optimal care is not predicted to vary directly and adaptively with $f_i$ or $f_j$ [31], but could potentially show inbreeding depression if inbred parents are resource-constrained. Consequently, $k_{ij}$ and $f_i$ or $f_j$, which constitute the fundamental underlying elements that determine parent–offspring relatedness and hence shape TAV and LAV, are predicted to have different direct effects on parental care [31]. However, the resulting suite of predictions regarding variation in parental care in relation to TAV, LAV, $k_{ij}$, $f_i$ and $f_j$ (table 1) has not been tested in any system.

We recorded rates at which adults provisioned broods of dependent offspring (hereafter 'feeding rates') as a measure of parental care in a song sparrow (Melospiza melodia) population where BS, EPP, $k_{ij}$, $f_i$ and $f_j$ and hence TAV and LAV vary substantially among individuals and breeding attempts [12,31], and tested three sets of hypotheses and associated predictions (table 1). First, we tested whether female and male feeding rates increased with increasing total relatedness to their brood, measured as $TAV_i$ or $TAV_j$. Since TAV is intrinsically positively correlated with BS overall but can vary within levels of BS (electronic supplementary material, appendix S1 and S4), we further tested whether feeding rates increased with TAV after controlling for BS. Second, we tested whether male feeding rates decreased with increasing LAV (or $P_{EPO}$), and hence with the value of offspring lost though EPP. Third, we focussed on the fundamental underlying elements and tested whether male and female feeding rates increased with increasing $k_{ij}$ but not with increasing $f_i$ or $f_j$ as predicted by inclusive fitness theory [31]. While the focal song sparrows are typically socially monogamous, some are socially polygynous (i.e. one male simultaneously socially paired with ≥ 2 females), and paternal care can be differentially allocated to offspring of different females [5,34–36]. We therefore additionally tested whether parental feeding rate varied with social status, and whether TAV, LAV, $k_{ij}$, $f_i$ and $f_j$ interacted with social status to shape patterns of parental care arising given complex reproductive interactions among relatives.

Proc. R. Soc. B 286: 20191933

**Table 1.** Summary of key focal variables and predictions based on underlying kin selection and inclusive fitness theory. Subscripts $i$ and $j$ refer to a socially paired male and female respectively, and $q$ refers to the female's extra-pair mate. Individuals $i$ and $j$ could produce within-pair offspring (WPO), while individuals $j$ and $q$ could produce extra-pair offspring (EPO) through extra-pair paternity (EPP). Full details of metric calculations are in electronic supplementary material, appendix S1.

| hypothesis set | focal variables | predicted response by males | predicted response by females |
| --- | --- | --- | --- |
| 1A | brood total allelic value (TAV) | paternal feeding rate will increase with increasing TAV more tightly than with increasing BS | maternal feeding rate will increase with increasing TAV more tightly than with increasing BS |
| 1B | brood total allelic value (TAV) controlling for brood size (BS) | paternal feeding rate will increase with increasing TAV after controlling for BS | maternal feeding rate will increase with increasing TAV after controlling for BS |
| 2 | lost allelic value (LAV) and paternity loss ($P_{EPO}$) | paternal feeding rate will decrease with increasing LAV and $P_{EPO}$ | maternal feeding rate will not vary directly with LAV or $P_{EPO}$ |
| 3A | coefficient of kinship between mates ($k_{ij}$) | paternal feeding rate will increase with increasing $k_{ij}$ | maternal feeding rate will increase with increasing $k_{ij}$ |
| 3B | individual's own coefficient of inbreeding ($f$) | paternal feeding rate will not vary with $f_i$ | maternal feeding rate will not vary with $f_j$ |

## 2. Methods

### (a) Study system

Testing the focal predictions (table 1) requires quantifying the degree of parental care expressed across family groups comprising social parents and WPO and/or EPO with known parental $k_{ij}$, $k_{iq}$, $k_{jq}$, $f_i$ and $f_j$. These data are available from a resident, pedigreed population of song sparrows on Mandarte Island, British Columbia, Canada [37].

On Mandarte, both song sparrow sexes can breed from age one year, and pairs typically rear 2–3 broods of 1–4 nestlings during April–July each year. Each year since 1975, all territories were mapped, all nests were monitored, and all nestlings and any immigrants were uniquely colour-ringed [35,36,38]. The socially paired adults attending each nest were identified and sexes were attributed from observed reproductive behaviour (male song, female incubation), allowing identification of socially monogamous and polygynous breeding pairs [35,36,38]. Genetic parentage analyses demonstrated 28% EPP (affecting 44% of broods), but no extra-pair maternity [39] (electronic supplementary material, appendix S2). Mandarte is part of a large meta-population [38] and the small local population size (mean 33.5 adult females, range 4–72), plus occasional immigrants (mean approx. 0.9/year) generates substantial variation in $k$ and $f$ [12].

### (b) Parental feeding rates

As a measure of parental care, we recorded parental feeding rates defined as the number of provisioning visits made to a focal nest per hour by each socially paired parent (electronic supplementary material, appendix S2). The dataset totalled 337 1-hour observation 'sessions' spanning the 12-day nestling period at 138 different nests (38, 46 and 44 in 2003, 2007 and 2008, respectively), with a median of 2 sessions per nest (range: 1–7). Nests attended by socially monogamous pairings were defined as 'monogamous' ($n = 79$). We defined each polygynous male's first hatched nest among broadly concurrent attempts as 'primary polygynous' ($n = 30$), and his second or third concurrent nest as 'secondary polygynous' ($n = 29$). Since females did not always pair with the same male across nesting attempts, and some females bred in multiple years, the 138 nests were attended by 65 and 54 different females and males, respectively (generating 75 different pairings).

### (c) Statistical analyses

We used standard pedigree algorithms to compute each individual's $f_i$ or $f_j$, and $k_{ij}$, $k_{iq}$ and $k_{jq}$ between individuals, and hence calculate relatedness between each focal parent and each nestling they reared. Male TAV ($TAV_i$), female TAV ($TAV_j$) and LAV were then calculated for each brood (electronic supplementary material, appendix S1). We fitted linear mixed-effects models (LMMs) to test specified hypotheses relating male and female feeding rates to the focal variables (table 1).

In general, feeding rates often vary with multiple non-focal variables, including nestling age [40,41], time of season and day [42], mate behaviour [43,44] and social status [35]. We therefore used a comparative modelling approach, and compared a null LMM that included baseline effects on feeding rate to LMMs that additionally included each focal variable. Baseline effects comprised nestling age (days after hatch, continuous variable), nest lay date (continuous variable), time of day (morning or afternoon, two-level factor) and nest social status (monogamous, primary polygynous or secondary polygynous three-level factor). Since effects on male and female feeding rates were modelled separately but experimental, empirical and theoretical studies suggest that a focal individual's behaviour might be influenced by its mate's behaviour [43,45,46], each null LMM also included the focal individual's mate's simultaneously observed feeding rate (as a continuous covariate) and interactions with social status. However, key model results remained quantitatively similar when mate feeding rate was removed.

First, to test the prediction that parental feeding rates increased with increasing TAV more than with increasing BS (table 1) we compared support for LMMs that additionally included TAV or BS versus the null LMM. Here, TAV and BS were modelled as continuous covariates, therefore adding one parameter to the null model. We then z-standardized TAV within each level of BS (i.e. $TAV_z = (TAV − \mu_{TAV})/\sigma_{TAV}$, where $\mu_{TAV}$ and $\sigma_{TAV}$ are the mean and standard deviation of TAV within each BS) and compared LMMs that included additive and interactive effects of $TAV_z$ and BS (as a four-level factor) to models that did not include $TAV_z$.

Second, to test the prediction that male but not female feeding rate decreased with increasing LAV (or $P_{EPO}$; table 1) we compared LMMs that additionally included each of these covariates to the null LMM. Third, to test the predictions that parental feeding rates would increase with increasing $k_{ij}$, but not vary

with $f_i$ and $f_j$, we compared LMMs that included each of these three covariates to the null LMM. These LMMs additionally included BS (as a continuous covariate). Finally, we expected male feeding rates to be lower at secondary polygynous nests, while female feeding rates could be higher if they compensated [43,44], implying that both sexes' feeding rates might depend on social status. Consequently, we additionally fitted LMMs that included 2-way interactions between nest social status and each focal variable.

LMMs assumed Gaussian distributions for feeding rates. All continuous variables within interaction terms were centred to minimize multicollinearity and aid model convergence. To account for non-independence across multiple observation sessions of the same nest and parents, random individual identity, social mate identity, and nest identity effects were included in all LMMs. We used Akaike information criterion, corrected for small sample sizes (AIC$_c$), to assess whether LMMs that included each focal predictor variable were better supported than the null LMM and/or than their competing predictor (e.g. TAV versus BS), defined as a difference in AIC$_c$ ($\Delta$AIC$_c$) equalling or exceeding two units [47].

All models were fitted using R 3.1.1 [48] with packages lme4 [49], lmerTest [50] and MuMIn [51]. Raw means are presented ± 1 s.d. Full distributions of all variables, and relationships between feeding rate and null variables, are in electronic supplementary material, appendix S3. LMM results are presented as standardized estimates (regression slope $\beta$) ± 1 s.e.). Estimates and SEs for factor levels not in interactions (i.e. brood size, time of day, and social status) are presented as least square means. Full details of all LMMs are in electronic supplementary material, appendix S6. Data are available from the Dryad Digital Repository: https://doi.org/10.5061/dryad.1zcrdfnf [52].

## 3. Results

### (a) Baseline effects of sex and social status

Across all observation sessions, mean female and male feeding rates were 6.4 ± 4.1 and 4.2 ± 3.4 trips per hour, respectively. Males had lower mean feeding rates at secondary polygynous nests (1.4 ± 2.3) than at primary polygynous (4.7 ± 3.8) or monogamous nests (5.1 ± 3.1), while females had higher mean feeding rates at secondary polygynous nests (9.1 ± 4.9) than at primary polygynous (6.4 ± 3.6) or monogamous nests (5.4 ± 3.4; electronic supplementary material, appendix S3 and S6). Male and female feeding rates were positively correlated at primary polygynous and monogamous nests (Pearson correlation coefficient: $r_p$ = 0.40, 0.44, respectively), but weakly negatively correlated at secondary polygynous nests ($r_p$ = −0.10; electronic supplementary material, appendix S3). Secondary females therefore partially compensated for lower feeding rates of their socially polygynous mates.

### (b) Brood size and total allelic value

Models for sex-specific feeding rates that additionally included BS (continuous variable) were substantially better supported than the null LMM for females ($\Delta$AIC$_c$ = −11.1), but only slightly better supported for males ($\Delta$AIC$_c$ = −0.7). These LMMs showed that feeding rate increased with increasing BS in females ($\beta$ = 0.87 ± 0.23), and tended to do so in males ($\beta$ = 0.38 ± 0.21; electronic supplementary material, appendix S3 and S6).

As expected, TAV$_i$ and TAV$_j$ were strongly but not perfectly positively correlated across the 139 observed broods ($r_p$ = 0.75), and TAV was positively correlated with BS in both sexes (males: $r_p$ = 0.54; females: $r_p$ = 0.76; electronic supplementary material, appendix S4 and S6). However, both TAV$_i$ and TAV$_j$ varied considerably within levels of BS, reflecting underlying variation in $P_{EPO}$, $k_{ij}$, $k_{iq}$, $k_{jq}$, $f_i$ and $f_j$ (electronic supplementary material, appendix S4). LMMs that included brood TAV were much better supported than the null LMM for both sexes (males: $\Delta$AIC$_c$ = −5.1, females: $\Delta$AIC$_c$ = −14.9), showing that feeding rate increased with TAV in both sexes (figure 1). Importantly, LMMs that included TAV were better supported than competing LMMs that included BS for males ($\Delta$AIC$_c$ = −4.3) and females ($\Delta$AIC$_c$ = −3.8; electronic supplementary material, appendix S6). Consequently, as predicted, male and female feeding rates were better explained by increasing TAV than by increasing BS.

Furthermore, models that included standardized TAV within brood size (TAV$_z$) were better supported than the null LMM (without BS) for females ($\Delta$AIC$_c$ = −2.9) and males ($\Delta$AIC$_c$ = −2.6). Models that included TAV$_z$ and BS were also better supported than the null with BS for females ($\Delta$AIC$_c$ = −3.0) and males ($\Delta$AIC$_c$ = −2.5; electronic supplementary material, appendix S6). Feeding rates, therefore, increased with TAV$_z$ within broods of each size (figure 1). LMMs that additionally included TAV$_z$ by BS interactions were marginally better supported than the null model (males: $\Delta$AIC$_c$ = −1.3, females: $\Delta$AIC$_c$ = −2.1), but less well supported than models without interactions (males: $\Delta$AIC$_c$ = 2.0, females: $\Delta$AIC$_c$ = 2.1). Meanwhile, LMMs that included TAV$_z$ by social status interactions were marginally better supported than the null model for males ($\Delta$AIC$_c$ = −1.7), and slightly less well supported for females ($\Delta$AIC$_c$ = +0.6). Overall, these results show that, in accordance with the prediction (table1), increased TAV was associated with increased parental feeding rates (figure 1).

### (c) Lost allelic value

Across all nests, $P_{EPO}$ varied between 0.00 and 1.00 (mean: 0.27 ± 0.35), and LAV varied between 0.000 and 1.913 (mean: 0.208 ± 0.373). As expected, LAV was positively correlated with $P_{EPO}$ across all 138 focal broods ($r_p$ = 0.89; electronic supplementary material, appendix S5). Yet some broods had low LAV relative to $P_{EPO}$, reflecting cases where cuckolded males were closely related to EPO (electronic supplementary material, appendix S5). $P_{EPO}$, and hence LAV, varied with social status. Specifically, primary polygynous nests had higher $P_{EPO}$ than monogamous nests (0.33 ± 0.30 versus 0.21 ± 0.30; $\beta$ = 0.12 ± 0.04, 85%CI: 0.06–0.19) and secondary polygynous nests (0.24 ± 0.30; $\beta$ = -0.10 ± 0.45, 85% CI: −0.19–0.004), while monogamous and secondary polygynous nests were similar ($\beta$ = 0.03 ± 0.45, 85%CI: −0.06–0.11; electronic supplementary material, appendix S6).

Models for sex-specific feeding rates that included LAV were slightly better supported than the null LMM (including BS as a covariate) for males ($\Delta$AIC$_c$ = −1.9), but less well supported for females ($\Delta$AIC$_c$ = +1.7). As predicted, male feeding rate tended to decrease with increasing LAV, but female feeding rate did not (figure 2). LMMs that additionally included LAV by social status interactions were slightly less well supported than the null LMM for males ($\Delta$AIC$_c$ = +1.0), but indicated that males at primary polygynous nests showed the greatest reduction in feeding rate with increasing LAV (figure 2; electronic supplementary material, appendix S6). There was no support for LMMs that included a LAV by social status interaction in females ($\Delta$AIC$_c$ = +7.9; figure 2;

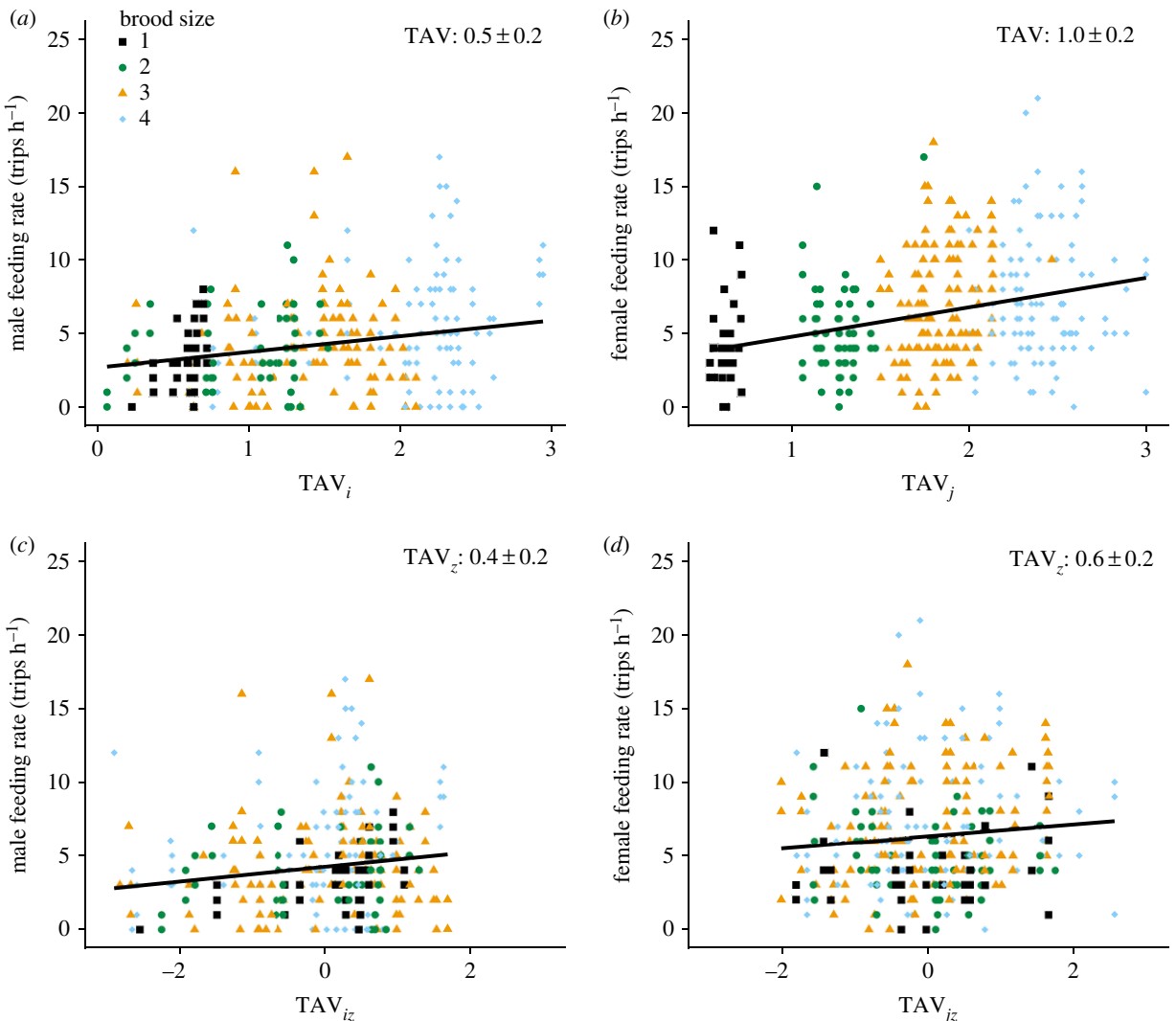

**Figure 1.** Relationships between male and female song sparrow parental feeding rates and (*a,b*) brood total allelic value (TAV$_i$ and TAV$_j$) and (*c,d*) standardized TAV within each level of brood size (TAV$_{iz}$ and TAV$_{jz}$). Points represent observation sessions. Colours and symbols denote different brood sizes (1: black, square; 2: green, large circle; 3: yellow, triangle; 4: blue, small circle). Lines show predicted regressions of feeding rates on TAV or TAV$_z$. Regression slopes are presented as $\beta$ estimates ± 1 s.e. (full details in electronic supplementary material, appendix S6). (Online version in colour.)

electronic supplementary material, appendix S6). Since LAV and $P_{EPO}$ were correlated, conclusions were very similar for models that included $P_{EPO}$ rather than LAV as the focal variable (electronic supplementary material, appendix S5 and S6).

### (d) Kinship (*k*) and inbreeding (*f*) coefficients

Individuals' coefficients of kinship with their social mates ($k_{ij}$) varied between 0.000 and 0.301 (mean: 0.087 ± 0.055). Models for sex-specific feeding rates that included $k_{ij}$ were better supported than the null LMM (including BS) for males ($\Delta AIC_c = -4.4$), but less well supported for females ($\Delta AIC_c = +2.0$; electronic supplementary material, appendix S6). Males in pairs with higher $k_{ij}$ had higher feeding rates but females did not (figure 3). LMMs that additionally included $k_{ij}$ by social status interactions were similarly supported as the null LMM for males ($\Delta AIC_c = -0.7$), but suggest that feeding rate tended to increase most markedly with increasing $k_{ij}$ at primary polygynous nests (figure 3). Such models were less well supported for females ($\Delta AIC_c = +1.8$), but suggest that females at secondary polygynous nests had lower feeding rates increasing $k_{ij}$ (figure 3).

Individuals' coefficients of inbreeding (*f*) varied between 0 and 0.164 (mean: 0.057 ± 0.035) for males and 0 and 0.181

(mean: 0.057 ± 0.039) for females. LMMs that included $f_i$ were marginally less well supported than the null LMM (including BS) for males ($\Delta AIC_c = +1.0$), and females ($\Delta AIC_c = +1.5$; electronic supplementary material, appendix S6). Overall, feeding rates did not vary markedly with $f_i$ in either sex (figure 3). However, LMMs that additionally included $f_i$ by social status interactions were slightly better supported than the null LMM in males ($\Delta AIC_c = -1.6$) but not females ($\Delta AIC_c = +4.3$). Male feeding rates tended to increase with increasing $f_i$ at primary polygynous nests, decrease with increasing $f_i$ at secondary polygynous nests, and did not vary with $f_i$ at monogamous nests (figure 3). Such patterns were not evident in females across social statuses (figure 3).

## 4. Discussion

Patterns of variation in parental feeding rates observed in song sparrows experiencing considerable natural variation in parent–brood relatedness, resulting from combinations of EPP, mate kinship and individual coefficient of inbreeding, broadly concurred with key predictions of inclusive fitness theory (table 1). A key result is that feeding rates of both sexes increased with increasing TAV of the dependent

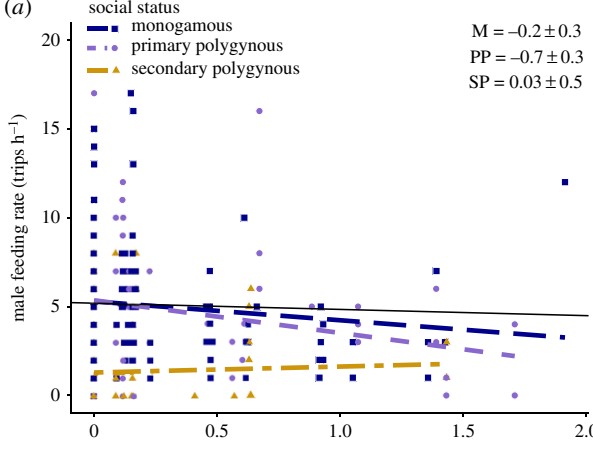

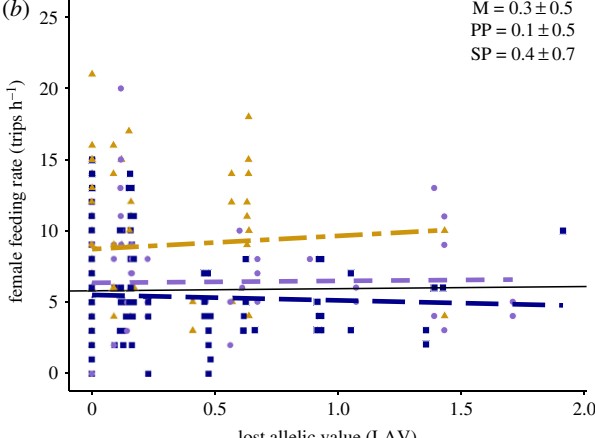

**Figure 2.** Relationships between (*a*) male and (*b*) female song sparrow parental feeding rates and lost allelic value (LAV). Colours and symbols indicate nest social status (monogamous (M): blue, square; primary polygynous (PP): purple, diamond; secondary polygynous (SP): yellow, triangle). Points represent observation sessions. Lines show predicted regressions of feeding rate on LAV overall (black), and for each social status. Regression slopes are presented as standardized $\beta$ estimates $\pm$ 1 s.e. from models that included a standardized LAV by social status interaction and represent the absolute slope (non-contrast) of the relationship. *Y*-axes are on different scales for males and females. (Online version in colour.)

brood, even after controlling for brood size (TAV$_z$; figure 1). Males and females consequently fed broods more often per hour as the expected number of identical-by-descent allele copies increased, to degrees that would generate notable increases in the total feeds received by highly related broods over the full nesting period (figure 1). These results support the central premise of existing models of optimal parental effort and investment that consider brood size [53] and relatedness [25], but provide conceptual and empirical advances by encompassing complex variation in relatedness arising from reproductive interactions among relatives [12].

Variation in brood TAV from the perspective of a potentially caring male partly reflects variation in paternity loss (proportion of offspring that are extra-pair; $P_{EPO}$) and kinship with his socially paired female ($k_{ij}$) and her extra-pair male(s) ($k_{iq}$) and resulting LAV. Our analyses provide some support for the prediction that male feeding rate will decrease with increasing LAV, and with increasing $P_{EPO}$ itself, and hence that males that lose relatedness to a dependent brood due to EPP provide less care. This concurs with some [6,13–15,23,54], but not all [6,15,54], previous empirical studies that tested whether paternal care decreases with increasing $P_{EPO}$.

However, our results highlight that key patterns of variation in paternal care might also depend on the social mating system. In particular, the negative effect of LAV on male feeding rate tended to be strongest at primary polygynous nests, perhaps reflecting the higher mean $P_{EPO}$ in these nests. Overall, these results support the hypothesis that female extra-pair reproduction can incur a cost in the form of reduced paternal care, potentially selecting against underlying polyandry [16,36], but imply that such costs might depend on social status [35].

Variation in TAV also reflects variation in kinship between socially paired mates ($k_{ij}$), and inclusive fitness theory predicts that optimal paternal care, interpreted as a component of parental investment, should increase with increasing $k_{ij}$ [31]. Our results strongly support this prediction for males (figure 3), translating into substantial increases in the number of paternal feeds received by inbred broods. Such increases might yield an evolutionary benefit of inbreeding, or at least negate the underlying evolutionary cost [55]. This is because inbreeding increases parent–offspring relatedness and hence propagation of identical-by-descent allele copies (given no 'opportunity cost' of lost outbred matings), but may also cause inbreeding depression in resulting offspring [32]. However, this cost can be negated if inbreeding parents can ameliorate inbreeding depression in resulting offspring through increased parental care [56]. This may be the case for song sparrows, since inbreeding depression in nestling survival from hatching to independence from parental care is weak [57], and inbreeding parents rear larger broods [58]. Consequently, there is weak selection against inbreeding despite strong inbreeding depression in individual fitness, and no evidence for active inbreeding avoidance through either social pairing or extra-pair reproduction [57,59]. By contrast, female song sparrows tended to decrease their feeding rate with increasing $k_{ij}$, perhaps reflecting a response to substantially increased male feeding rate. But, generally, it is unclear if increased levels of male care would be strong enough to substantially decrease inbreeding depression. Indeed, additional male care did not decrease inbreeding depression in burying beetles (*Nicrophorus vespilloides*) [60]. However, our results add to several recent experimental studies on diverse taxa suggesting parents mated to kin may adjust reproductive strategies to reduce inbreeding depression, for example by reducing their clutch size in burying beetles [61], gaining alloparental care from helpers in red-winged fairy-wrens (*Malurus elegans*) [62], providing increased levels of prenatal maternal provisioning in Japanese quail (*Coturnix japonica*) [63], adopting group living and maternal care in social spiders (*Anelosimus* cf. *jucundus*) [64], or more cooperative parental behaviour in an African cichlid (*Pelvicachromis taeniatus*) [28]. Yet, to our knowledge, no previous studies have directly examined how mate kinship influences parental care in wild non-cooperative breeding species. Since inbreeding occurs in many species (e.g. [27,56,63,65]) such effects warrant wider attention in the context of inclusive fitness theory [1]. In our system, and others, this could potentially include examining female responses to kinship with their extra-pair mates ($k_{jq}$).

Our results also broadly concur with the prediction that overall feeding rates should not vary with a parent's own $f_i$, insofar as such null predictions can be rigorously tested. These results can also be interpreted to provide no overall evidence of direct inbreeding depression in parental feeding rates. The few previous studies quantifying inbreeding effects

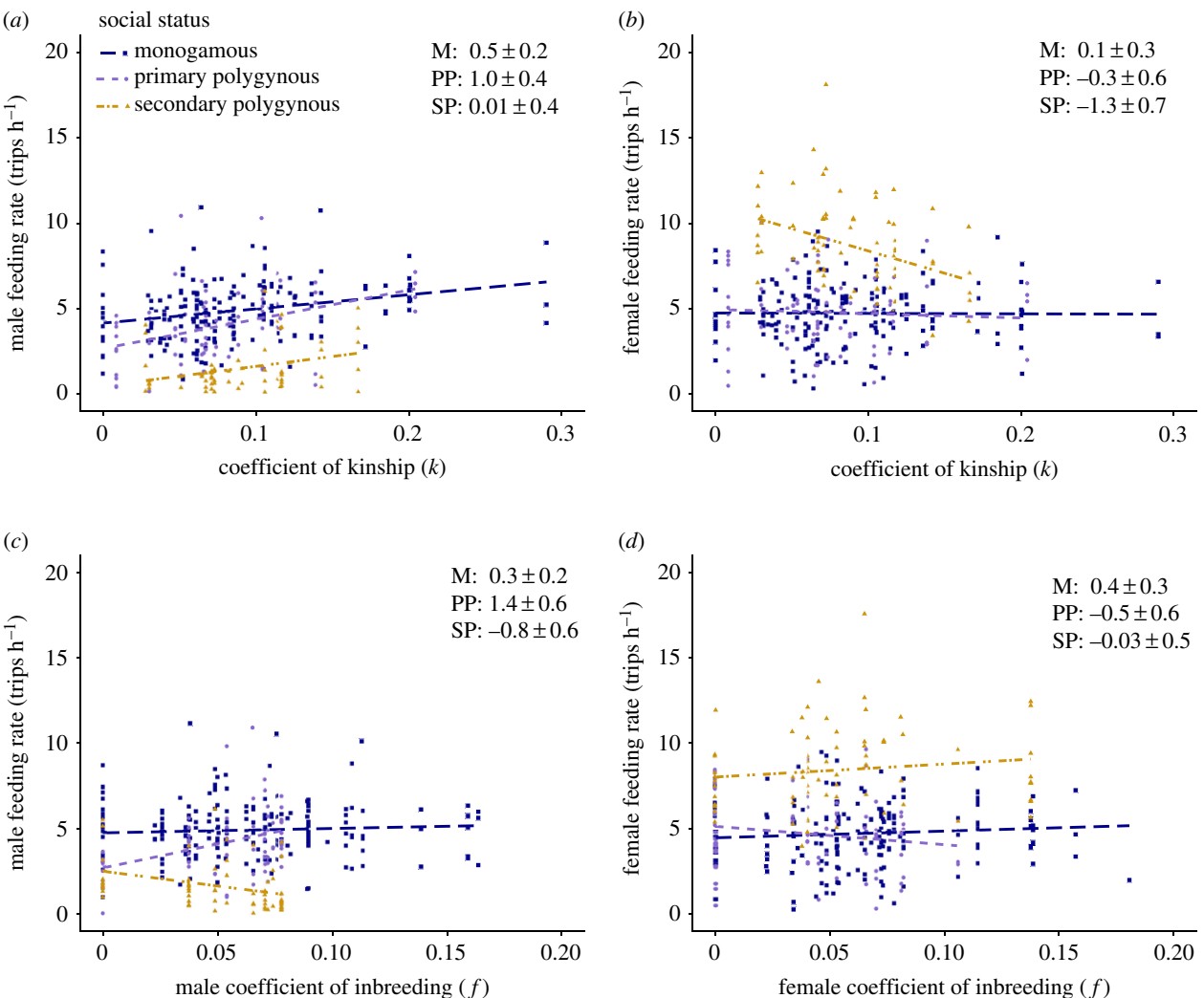

**Figure 3.** Relationships between male and female song sparrow parental feeding rates and (*a,b*) pair coefficient of kinship ($k_{ij}$) and (*c,d*) individual coefficient of inbreeding ($f_i$ or $f_j$). Colours and symbols indicate nest social status (monogamous (M): blue, square; primary polygynous (PP): purple, diamond; secondary polygynous (SP): yellow, triangle). Points represent observation sessions. Lines show predicted regressions of feeding rate on $k_{ij}$, $f_i$ or $f_j$. Regression slopes are presented as standardized $\beta$ estimates $\pm$ 1 s.e. from LMMs that included a standardized focal variable by social status interaction and represent the absolute slope (non-contrast) of the relationship. *y*-axes are on different scales for males and female. (Online version in colour.)

on parental care all compared highly inbred (e.g. $f \geq 0.25$) to outbred parents in captivity. Inbred versus outbred prairie voles (*Microtus ochrogaster*) and burying beetles did not differ in multiple parental behaviours [66,67], whereas inbred female zebra finches (*Taeniopygia guttata*) incubated less than outbred females [68]. However, in song sparrows, the effects of $f_i$ on parental feeding rate appear to vary strongly with social status: male feeding rate increased markedly with increasing $f_i$ at primary polygynous nests but decreased at secondary polygynous nests, perhaps reflecting re-allocation of parental investment among broods by more inbred males. Future studies should further examine how effects of $f$ on key parental behaviours are shaped by the social mating system.

Parental feeding rate is one key component of parental care that may be positively or negatively correlated with other components. Consequently, the degree to which variation in feeding rate captures variation in overall care, or in parental investment strictly defined [5,34], is unknown. Nevertheless, our results are striking in showing that one major component of care does vary with subtle variation in relatedness in accordance with inclusive fitness theory (table 1), especially in males. This raises interesting questions regarding how such outcomes could arise. Our results are

inevitably correlative and hence cannot prove causal effects; but any experimental manipulation of such effects in free-living populations would be exceptionally challenging, and our analyses controlled for key potentially confounding variables that are known to affect feeding rates. The observed increases in parental feeding rates with increasing $TAV_z$ may therefore imply that song sparrows can respond to direct or indirect cues of relatedness. Some mechanisms by which this could be achieved have previously been identified in song sparrows. Specifically, preen wax composition, male song repertoire size and demographic status have been shown to indicate relatedness [69–71], but, of course, other mechanisms, such as differential offspring behaviour, might also be involved.

Ethics. This research was approved by the University of British Columbia's Animal Care Committee.

Data accessibility. Data available from the Dryad Digital Repository: https://doi.org/10.5061/dryad.1zcrjdfnf [52]. R code supporting this article has been uploaded as part of the electronic supplementary material.

Authors' contributions. E.A.G. and J.M.R. designed the research and wrote the manuscript. E.A.G. analysed the data. All other authors conducted key fieldwork and contributed to manuscript editing.

**Competing interests.** We declare no competing interests.

**Funding.** National Sciences and Engineering Research Council (P.A. and E.A.G); Izaak Walton Killam Memorial Fund for Advanced Studies (E.A.G. and J.M.R.), UK Natural Environment Research Council (R.J.S.) and the European Research Council (J.M.R.) provided funding.

**Acknowledgements.** We thank the Tsawout and Tseycum First Nation bands for allowing access to Mandarte, numerous field assistants, graduate students and postdoctoral fellows who contributed to long-term data collection and Brad Duthie for insightful discussions regarding underlying concepts.

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
