## [Reviewer comments · Proceedings of the Royal Society B: Biological Sciences]

Review History

RSPB-2019-1933.R0 (Original submission)

Review form: Reviewer 1

Recommendation

Major revision is needed (please make suggestions in comments)

Scientific importance: Is the manuscript an original and important contribution to its field?

Excellent

General interest: Is the paper of sufficient general interest?

Good

Quality of the paper: Is the overall quality of the paper suitable?

Good

Is the length of the paper justified?

Yes

Should the paper be seen by a specialist statistical reviewer?

No

Do you have any concerns about statistical analyses in this paper? If so, please specify them explicitly in your report.

Yes

It is a condition of publication that authors make their supporting data, code and materials available - either as supplementary material or hosted in an external repository. Please rate, if applicable, the supporting data on the following criteria.

Is it accessible?

Yes

Is it clear?

Yes

Is it adequate?

Yes

Do you have any ethical concerns with this paper?

No

Comments to the Author

This manuscript by Gow et al. formulates several predictions of inclusive fitness theory about parental care behaviour. The authors test these predictions in a population of song sparrows characterised by social monogamy/polygyny and moderate levels of both inbreeding and extra-pair paternity. The authors provide a very fine-grained analysis of predicted care behaviour under varying levels of relatedness between parents of offspring due to (i) extra-pair paternity, (ii) relatedness between social mates, (iii) relatedness between socially paired and extra-pair fathers of a given brood, and (iv) inbreeding in the previous generation, which tends to increase an individual's relatedness to its own offspring. Using a pedigree, they estimate the 'total allelic value' (TAV) of a brood to each parent, as well as the lost allelic value to males due to extra-pair paternity.

The authors' main predictions are that (i) male feeding rates should increase with TAV more tightly than with brood size; (ii) male (but not female) feeding rates should decrease with lost paternity; (iii) that feeding rates should increase with relatedness between social mates, but not with individuals' own coefficient of inbreeding. They found good support for the first prediction, and mixed support for the other two.

I found this an extremely well-written manuscript and appreciated the authors' commitment to formulating and testing precise hypotheses. I also liked the inclusion of scatterplots as a 'see for yourself' to complement the fancy statistical models. I think this study will make an excellent contribution to the literature. I do have a few concerns, however.

1. Definition of PKD: I was confused about the definition of the 'paternal kinship difference' (PKD). In the introduction, this is defined simply as $PKD = PAV - TAV$, where PAV is the total allelic value of the brood if the socially paired male had fathered all offspring, and TAV is the actual allelic value given his realised paternity. So under this definition, PKD should be in units of gene copies. So far so good.

But on lines 279-80, I was confused why PKD should be so tightly correlated with the proportion of extra-pair paternity, P_EPO ($r = 0.94$). Shouldn't variation in brood size weaken this correlation substantially? Is it possible that for this correlation you used something like PKD/(brood size) rather than absolute PKD? Brood size variation in your dataset is certainly not negligible.

Similarly, on lines 281-284, you say that P_EPO ranged from 0 to 1 (with mean of 0.27), whereas PKD ranged from 0 to 0.48. The latter range seems very low if PKD is measured in units of gene copies. For a brood with four chicks, the PAV to a male with full paternity would be around 2 (ignoring the second-order effects of individual inbreeding coefficients and relatedness among mates and competitors). Extra-pair paternity would need to be very high to reduce this to 0.48. Did all the broods with 3 or 4 chicks have very low P_EPO? None of the numbers you present seem impossible, just unlikely.

2. The prediction that male feeding rates should decrease with increase PKD is a bit weird, as brood size is an important confounder. If brood size is held constant, then this prediction makes perfect sense. This problem does not arise with P_EPO, which is expressed as a proportion (but perhaps your PKD is as well? See previous comment.)

3. Analysis of small AIC values: The manuscript relies heavily on the Akaike Information Criterion to compare models. The authors use a cut-off of $\Delta AIC_c \geq 2$ to determine whether two models differ substantially in fit. This is already a fairly small (though commonly used) cut-off, but the authors often sneak in interpretations of delta-values that are even smaller than this, where I would think it wiser not to interpret at all. Some examples are on lines 269-171, 310-311, 344-348.

4. Terminology and notation: The terms TAV, PAV, and PKD are all in the same currency of expected gene copies (I think, see comment 1). Maybe it would make more sense to rename PKD 'lost allelic value' (LAV) to make this connection clearer? Also perhaps 'total allelic value' could be renamed 'realised allelic value' (RAV) or similar. These are just suggestions.

I found the term 'total relatedness' (e.g. lines 83, 331) less than helpful, as it suggests the probability of common descent rather than the number of gene copies. You use it as a synonym for 'total allelic value', which I think is a better term.

Some minor suggestions:

Lines 83, 87: The period for multiplication is not very standard. Please change it to a vertically centred dot (at proofs stage if not earlier).

Line 92: Perhaps 'multiple' could be deleted?

Line 95-107: To me it would be more intuitive to first explain relatedness to WPO, then relatedness to EPO.

Line 116: suggest changing 'as if' -> 'if'

Lines 144, 207: It would help to explain why you don't expect feeding rates to depend on individuals' inbreeding coefficients. Is it because these coefficients do not differ among an individual's various brood over their lifetime?

Lines 172-178: I can't get the numbers here to add up. Some additional explanation would help. For instance, why are there 139 nests but only 82 females? Is it because the same female can have

a nest in multiple years? Do females ever have more than one nest in a single year? Also, you classify nests as monogamous/primary polygynous/secondary polygynous, but the numbers of each type of nest do not add up to the total number of nests (i.e. $82 + 30 + 29$ does not equal 139). Also, why is the number of monogamous nests the same as the number of females?

Lines 194, 209: Saying that an LMM 'includes' a regression on a particular parameter is a bit clumsy. The LMM is a regression. It would be nicer to say that the parameter 'was included as a covariate'.

Line 195: What happens if you leave the effect of the social partner's feeding rate out of these models? One could argue that this you are partially controlling for a consequence of the dependent variable (although the reality is probably more nuanced).

Lines 222-225: It's not clear to me where these reference values come from.

Line 243: The phrase 'across sessions' is a bit confusing, as at first I thought you were talking about autocorrelations of a single individual's feeding rate across different sessions. I think you are talking about the correlation between male and female feeding rate at a given nest. Maybe just delete these two words?

Line 255: suggest changing 'completely' -> 'perfectly'

Lines 250-264: I couldn't get the delta AICc values for females to add up. Write A0, A1 and A2 for the absolute AICc values for the null model, the model with BS only, and the model with TAV only. Then I think you are saying that:

$$A1 - A0 = -12.1$$

$$A2 - A0 = -14.9$$

$$A2 - A1 = -5.0$$

But these equations are inconsistent (e.g. subtracting the first equation from the second gives $A2 - A1 = -2.8$). What's going on here?

Lines 265-268: I don't understand the need for the model without BS. Since BS is informative and presumably not highly correlated with TAVz, why not always include it?

Lines 285-6: The differences between these values don't look as though they would be significant. Are they?

Lines 327-9: But for males the model with TAVz and BS fit slightly worse than the model with BS alone (line 268), seemingly contradicting this claim? I think your results broadly support the underlying prediction, but maybe this sentence needs to be re-worded.

Lines 366-8: delete repeated phrase

Fig 2 caption: Please include reference to the lower bound of zero for PKD.

Review form: Reviewer 2

Recommendation

Accept with minor revision (please list in comments)

Scientific importance: Is the manuscript an original and important contribution to its field?
Excellent

General interest: Is the paper of sufficient general interest?
Excellent

Quality of the paper: Is the overall quality of the paper suitable?
Excellent

Is the length of the paper justified?
Yes

Should the paper be seen by a specialist statistical reviewer?
No

Do you have any concerns about statistical analyses in this paper? If so, please specify them explicitly in your report.
No

It is a condition of publication that authors make their supporting data, code and materials available - either as supplementary material or hosted in an external repository. Please rate, if applicable, the supporting data on the following criteria.

Is it accessible?
Yes

Is it clear?
Yes

Is it adequate?
Yes

Do you have any ethical concerns with this paper?
No

Comments to the Author

This is a well conceived and well executed study relating offspring-parent genetic relatedness (which can be different for males and females due to extra-pair paternity), within-breeding pair relatedness and parental inbreeding coefficient to parental provisioning (i.e. feeding rate) by using a well established biparental bird system. This study adds a very interesting facet to the already impressive work of the group. The design is clear and well thought and the data seem to be analyzed appropriately. The MS is very well written. As predicted by kin selection theory, feeding rate of both sexes increases with increasing relatedness to the offspring but increasing within-pair relatedness increases feeding rates of males only. The results are very interesting and should appeal a broad readership.

Comments

-What could be the underlying kin recognition mechanism allowing such fine-tuned differentiations (especially when taken into account multiple paternities): maternal imprinting, self-reference? Does offspring behavior play a role, e.g. half sibling broods might differ from full-sibling broods and that could be used as proxy for paternal relatedness.
-Does variation in quality (e.g. body size) within and among pairs play a role? Quality assortative mating might increase within-pair genetic relatedness.

-How reliable is the feeding rate? Is it a reliable predictor of the amount of food provided or is it prone to cheating?

-This study is not the first one that shows the interrelation between inbreeding, genetic relatedness of parents and parental brood care. In 2007 Thünken et al. published the experimental study "Active inbreeding in a cichlid fish and its adaptive significance" (Current Biology) in which the authors showed kin mating preferences in a biparental cichlid fish and proposed a potential benefit of inbreeding that had not been addressed so far: that the higher genetic relatedness between parents might improve parental care because of a reduced sexual conflict over care due to kin selection. In accordance with this idea the study also showed that related parents, i.e. full siblings, spent more time protecting the young than unrelated parents and that males were less aggressive when the female partner was a relative. Although the paper is mentioned as reference for active inbreeding (lines 93 and 376), its pioneering aspect (link between inbreeding, kin selection and parental care) relevant for the present study should be acknowledged.

-the high number of abbreviations makes it a bit difficult to read the ms, especially the discussion. Maybe add a table summarizing the abbreviation.

Decision letter (RSPB-2019-1933.R0)

18-Sep-2019

Dear Dr Gow:

Your manuscript has now been peer reviewed and the reviews have been assessed by an Associate Editor. The reviewers, AE, and I all find your manuscript to be a well written and a strong test of inclusive fitness theory. However, both reviewers and the AE raise a number of suggestions to strengthen and clarify your manuscript.

The reviewers' comments (not including confidential comments to the Editor) and the comments from the Associate Editor are included at the end of this email for your reference. I invite you to revise your manuscript in light of these suggestions.

Research ethics:

Use of animals and field studies:

Please submit a copy of your revised paper within three weeks. If we do not hear from you

within this time your manuscript will be rejected. If you are unable to meet this deadline please let us know as soon as possible, as we may be able to grant a short extension.

Best wishes,
Dr Sarah Brosnan
mailto: proceedingsb@royalsociety.org

Associate Editor

Comments to Author:

We have now received reports from two expert reviewers. Both commend on the high quality of the study and its presentation and recommend the manuscript for publication, provided their comments will be addressed in a revision. Please also consider two minor comments from my part:

L83: Relatedness is typically interpreted as the proportion of IBD alleles between two individuals (with a maximum value of 1). Therefore, a quantity composed of relatedness * brood size * proportion of paternity (that is, a sum of relatedness coefficients across parent-offspring pairs) should not be called relatedness, but something else. Later, this is defined as TAV, and this variable could be introduced here already.

L96: an example for relatedness between nesting males and the EPMs of their mates in a natural population of a biparental breeder is reported by Bose et al. 2019, BMC Biology 17:2.

Reviewer(s)' Comments to Author:

Referee: 1

Comments to the Author(s)

This manuscript by Gow et al. formulates several predictions of inclusive fitness theory about parental care behaviour. The authors test these predictions in a population of song sparrows characterised by social monogamy/polygyny and moderate levels of both inbreeding and extra-pair paternity. The authors provide a very fine-grained analysis of predicted care behaviour under varying levels of relatedness between parents of offspring due to (i) extra-pair paternity, (ii) relatedness between social mates, (iii) relatedness between socially paired and extra-pair fathers of a given brood, and (iv) inbreeding in the previous generation, which tends to increase an individual's relatedness to its own offspring. Using a pedigree, they estimate the 'total allelic value' (TAV) of a brood to each parent, as well as the lost allelic value to males due to extra-pair paternity.

The authors' main predictions are that (i) male feeding rates should increase with TAV more tightly than with brood size; (ii) male (but not female) feeding rates should decrease with lost paternity; (iii) that feeding rates should increase with relatedness between social mates, but not with individuals' own coefficient of inbreeding. They found good support for the first prediction, and mixed support for the other two.

I found this an extremely well-written manuscript and appreciated the authors' commitment to formulating and testing precise hypotheses. I also liked the inclusion of scatterplots as a 'see for yourself' to complement the fancy statistical models. I think this study will make an excellent contribution to the literature. I do have a few concerns, however.

1. Definition of PKD: I was confused about the definition of the 'paternal kinship difference' (PKD). In the introduction, this is defined simply as $PKD = PAV - TAV$, where PAV is the total

allelic value of the brood if the socially paired male had fathered all offspring, and TAV is the actual allelic value given his realised paternity. So under this definition, PKD should be in units of gene copies. So far so good.

But on lines 279-80, I was confused why PKD should be so tightly correlated with the proportion of extra-pair paternity, P_EPO ($r = 0.94$). Shouldn't variation in brood size weaken this correlation substantially? Is it possible that for this correlation you used something like PKD/(brood size) rather than absolute PKD? Brood size variation in your dataset is certainly not negligible.

Similarly, on lines 281-284, you say that P_EPO ranged from 0 to 1 (with mean of 0.27), whereas PKD ranged from 0 to 0.48. The latter range seems very low if PKD is measured in units of gene copies. For a brood with four chicks, the PAV to a male with full paternity would be around 2 (ignoring the second-order effects of individual inbreeding coefficients and relatedness among mates and competitors). Extra-pair paternity would need to be very high to reduce this to 0.48. Did all the broods with 3 or 4 chicks have very low P_EPO? None of the numbers you present seem impossible, just unlikely.

2. The prediction that male feeding rates should decrease with increase PKD is a bit weird, as brood size is an important confounder. If brood size is held constant, then this prediction makes perfect sense. This problem does not arise with P_EPO, which is expressed as a proportion (but perhaps your PKD is as well? See previous comment.)

3. Analysis of small AIC values: The manuscript relies heavily on the Akaike Information Criterion to compare models. The authors use a cut-off of $\Delta AIC_c \geq 2$ to determine whether two models differ substantially in fit. This is already a fairly small (though commonly used) cut-off, but the authors often sneak in interpretations of delta-values that are even smaller than this, where I would think it wiser not to interpret at all. Some examples are on lines 269-171, 310-311, 344-348.

4. Terminology and notation: The terms TAV, PAV, and PKD are all in the same currency of expected gene copies (I think, see comment 1). Maybe it would make more sense to rename PKD 'lost allelic value' (LAV) to make this connection clearer? Also perhaps 'total allelic value' could be renamed 'realised allelic value' (RAV) or similar. These are just suggestions.

I found the term 'total relatedness' (e.g. lines 83, 331) less than helpful, as it suggests the probability of common descent rather than the number of gene copies. You use it as a synonym for 'total allelic value', which I think is a better term.

Some minor suggestions:

Lines 83, 87: The period for multiplication is not very standard. Please change it to a vertically centred dot (at proofs stage if not earlier).

Line 92: Perhaps 'multiple' could be deleted?

Line 95-107: To me it would be more intuitive to first explain relatedness to WPO, then relatedness to EPO.

Line 116: suggest changing 'as if' -> 'if'

Lines 144, 207: It would help to explain why you don't expect feeding rates to depend on individuals' inbreeding coefficients. Is it because these coefficients do not differ among an individual's various brood over their lifetime?

Lines 172-178: I can't get the numbers here to add up. Some additional explanation would help. For instance, why are there 139 nests but only 82 females? Is it because the same female can have a nest in multiple years? Do females ever have more than one nest in a single year? Also, you classify nests as monogamous/primary polygynous/secondary polygynous, but the numbers of each type of nest do not add up to the total number of nests (i.e. $82 + 30 + 29$ does not equal 139). Also, why is the number of monogamous nests the same as the number of females?

Lines 194, 209: Saying that an LMM 'includes' a regression on a particular parameter is a bit clumsy. The LMM is a regression. It would be nicer to say that the parameter 'was included as a covariate'.

Line 195: What happens if you leave the effect of the social partner's feeding rate out of these models? One could argue that this you are partially controlling for a consequence of the dependent variable (although the reality is probably more nuanced).

Lines 222-225: It's not clear to me where these reference values come from.

Line 243: The phrase 'across sessions' is a bit confusing, as at first I thought you were talking about autocorrelations of a single individual's feeding rate across different sessions. I think you are talking about the correlation between male and female feeding rate at a given nest. Maybe just delete these two words?

Line 255: suggest changing 'completely' -> 'perfectly'

Lines 250-264: I couldn't get the delta AICc values for females to add up. Write A0, A1 and A2 for the absolute AICc values for the null model, the model with BS only, and the model with TAV only. Then I think you are saying that:

$$A1 - A0 = -12.1$$

$$A2 - A0 = -14.9$$

$$A2 - A1 = -5.0$$

But these equations are inconsistent (e.g. subtracting the first equation from the second gives $A2 - A1 = -2.8$). What's going on here?

Lines 265-268: I don't understand the need for the model without BS. Since BS is informative and presumably not highly correlated with TAVz, why not always include it?

Lines 285-6: The differences between these values don't look as though they would be significant. Are they?

Lines 327-9: But for males the model with TAVz and BS fit slightly worse than the model with BS alone (line 268), seemingly contradicting this claim? I think your results broadly support the underlying prediction, but maybe this sentence needs to be re-worded.

Lines 366-8: delete repeated phrase

Fig 2 caption: Please include reference to the lower bound of zero for PKD.

Referee: 2

Comments to the Author(s)

This is a well conceived and well executed study relating offspring-parent genetic relatedness (which can be different for males and females due to extra-pair paternity), within-breeding pair

relatedness and parental inbreeding coefficient to parental provisioning (i.e. feeding rate) by using a well established biparental bird system. This study adds a very interesting facet to the already impressive work of the group. The design is clear and well thought and the data seem to be analyzed appropriately. The MS is very well written. As predicted by kin selection theory, feeding rate of both sexes increases with increasing relatedness to the offspring but increasing within-pair relatedness increases feeding rates of males only. The results are very interesting and should appeal a broad readership.

Comments

- What could be the underlying kin recognition mechanism allowing such fine-tuned differentiations (especially when taken into account multiple paternities): maternal imprinting, self-reference? Does offspring behavior play a role, e.g. half sibling broods might differ from full-sibling broods and that could be used as proxy for paternal relatedness.
- Does variation in quality (e.g. body size) within and among pairs play a role? Quality assortative mating might increase within-pair genetic relatedness.
- How reliable is the feeding rate? Is it a reliable predictor of the amount of food provided or is it prone to cheating?
- This study is not the first one that shows the interrelation between inbreeding, genetic relatedness of parents and parental brood care. In 2007 Thünken et al. published the experimental study "Active inbreeding in a cichlid fish and its adaptive significance" (Current Biology) in which the authors showed kin mating preferences in a biparental cichlid fish and proposed a potential benefit of inbreeding that had not been addressed so far: that the higher genetic relatedness between parents might improve parental care because of a reduced sexual conflict over care due to kin selection. In accordance with this idea the study also showed that related parents, i.e. full siblings, spent more time protecting the young than unrelated parents and that males were less aggressive when the female partner was a relative. Although the paper is mentioned as reference for active inbreeding (lines 93 and 376), its pioneering aspect (link between inbreeding, kin selection and parental care) relevant for the present study should be acknowledged.
- the high number of abbreviations makes it a bit difficult to read the ms, especially the discussion. Maybe add a table summarizing the abbreviation.

Author's Response to Decision Letter for (RSPB-2019-1933.R0)

See Appendix A.

RSPB-2019-1933.R1 (Revision)

Review form: Reviewer 1

Recommendation

Accept as is

Scientific importance: Is the manuscript an original and important contribution to its field?

Good

General interest: Is the paper of sufficient general interest?

Good

Quality of the paper: Is the overall quality of the paper suitable?

Excellent

Is the length of the paper justified?

Yes

Should the paper be seen by a specialist statistical reviewer?

No

Do you have any concerns about statistical analyses in this paper? If so, please specify them explicitly in your report.

No

It is a condition of publication that authors make their supporting data, code and materials available - either as supplementary material or hosted in an external repository. Please rate, if applicable, the supporting data on the following criteria.

Is it accessible?

No

Is it clear?

Yes

Is it adequate?

Yes

Do you have any ethical concerns with this paper?

No

Comments to the Author

I reviewed this manuscript once previously (Referee 1). I thank the authors for their revisions, which have cleared up many small discrepancies and confusions from the previous submission. I think this study will make a very nice contribution to the literature. I have only one further comment, which is to note the non-standard use of 85% confidence intervals. I'm not sure what Proc B's policy is on this kind of thing, so I will leave that up to the editor.

Decision letter (RSPB-2019-1933.R1)

06-Nov-2019

Dear Dr Gow

I am pleased to inform you that your manuscript entitled "Testing predictions of inclusive fitness theory in inbreeding relatives with biparental care" has been accepted for publication in Proceedings B.

You can expect to receive a proof of your article from our Production office in due course, please check your spam filter if you do not receive it. PLEASE NOTE: you will be given the exact page

length of your paper which may be different from the estimation from Editorial and you may be asked to reduce your paper if it goes over the 10 page limit.

Open Access

Paper charges

Sincerely,

Dr Sarah Brosnan
Editor, Proceedings B
<mailto:proceedingsb@royalsociety.org>

Associate Editor:

Board Member: 1

Comments to Author:

(There are no comments.)

Board Member: 2

Comments to Author:

(There are no comments.)

Appendix A

Dear Dr. Brosnan,

Thank you very much for your comments, and those of the Associate Editor and Reviewers, on our manuscript entitled: “Testing predictions of inclusive fitness theory in inbreeding relatives with biparental care” (ID PSPB-2019-1933). We were very pleased to receive such positive and helpful comments.

We have fully addressed all of the Associate Editor’s and Reviewers’ comments, most of which required minor edits and clarifications to the text. Full details of the changes are below. Please note that line numbers refer to the line numbers in the track changes version of the manuscript. The italics are the reviewer comments, and our responses are indicated below.

Response to Referees: “Testing predictions of inclusive fitness theory in inbreeding relatives with biparental care” RSPB-2019-1933

Response to Associate Editor:

We have now received reports from two expert reviewers. Both commend on the high quality of the study and its presentation and recommend the manuscript for publication, provided their comments will be addressed in a revision.

Author Response: Thank you very much for your positive comments. We believe we have been able to fully address the reviewers’ comments, as explained below.

*L83: Relatedness is typically interpreted as the proportion of IBD alleles between two individuals (with a maximum value of 1). Therefore, a quantity composed of relatedness * brood size * proportion of paternity (that is, a sum of relatedness coefficients across parent-offspring pairs) should not be called relatedness, but something else. Later, this is defined as TAV, and this variable could be introduced here already.*

Author response: We have now defined TAV at the first mention of brood relatedness, as suggested (Line 74–76).

L96: an example for relatedness between nesting males and the EPMs of their mates in a natural population of a biparental breeder is reported by Bose et al. 2019, BMC Biology 17:2.

Author response: Thank you for the suggestion to consider this recent relevant paper. We have now cited it as suggested (Line 101).

Response to Referee 1:

Comments to the Author(s)

This manuscript by Gow et al. formulates several predictions of inclusive fitness theory about parental care behaviour. The authors test these predictions in a population of song sparrows characterised by social monogamy/polygyny and moderate levels of both inbreeding and extra-pair paternity. The authors provide a very fine-grained analysis of predicted care behaviour under varying levels of relatedness between parents of offspring due to (i) extra-pair paternity, (ii) relatedness between social mates, (iii) relatedness between socially paired and extra-pair fathers of a given brood, and (iv) inbreeding in the previous generation, which tends to increase an individual's relatedness to its own offspring. Using a pedigree, they estimate the 'total allelic

value' (TAV) of a brood to each parent, as well as the lost allelic value to males due to extra-pair paternity.

The authors' main predictions are that (i) male feeding rates should increase with TAV more tightly than with brood size; (ii) male (but not female) feeding rates should decrease with lost paternity; (iii) that feeding rates should increase with relatedness between social mates, but not with individuals' own coefficient of inbreeding. They found good support for the first prediction, and mixed support for the other two.

I found this an extremely well-written manuscript and appreciated the authors' commitment to formulating and testing precise hypotheses. I also liked the inclusion of scatterplots as a 'see for yourself' to complement the fancy statistical models. I think this study will make an excellent contribution to the literature. I do have a few concerns, however.

Author response: Thank you very much for the your positive comments about our manuscript and for your thorough and helpful review. We have revised our manuscript accordingly, as explained below.

1. Definition of PKD: I was confused about the definition of the 'paternal kinship difference' (PKD). In the introduction, this is defined simply as $PKD = PAV - TAV$, where PAV is the total allelic value of the brood if the socially paired male had fathered all offspring, and TAV is the actual allelic value given his realised paternity. So under this definition, PKD should be in units of gene copies. So far so good.

But on lines 279-80, I was confused why PKD should be so tightly correlated with the proportion of extra-pair paternity, P_EPO ($r = 0.94$). Shouldn't variation in brood size weaken this correlation substantially? Is it possible that for this correlation you used something like $PKD/(brood\ size)$ rather than absolute PKD? Brood size variation in your dataset is certainly not negligible.

Similarly, on lines 281-284, you say that P_EPO ranged from 0 to 1 (with mean of 0.27), whereas PKD ranged from 0 to 0.48. The latter range seems very low if PKD is measured in units of gene copies. For a brood with four chicks, the PAV to a male with full paternity would be around 2 (ignoring the second-order effects of individual inbreeding coefficients and relatedness among mates and competitors). Extra-pair paternity would need to be very high to reduce this to 0.48. Did all the broods with 3 or 4 chicks have very low P_EPO ? None of the numbers you present seem impossible, just unlikely.

Author response: Thank you very much for your thoroughness, we checked our calculations for PKD (now termed 'LAV', see below) and noticed a scaling error, which we have now fixed. This change fully resolved the queries about the numerical values of PKD, but did not alter the key results of our analyses, and hence our main conclusions.

We have now made changes to Lines: 347–349, 356–358, 360, 437. We also updated Table S3, revised Figure 2, and updated the Supplement (Lines 198–199), Figure S4 and S11. We made no changes to the Discussion, Introduction or Methods, since these were correct in the original manuscript.

2. The prediction that male feeding rates should decrease with increase PKD is a bit weird, as brood size is an important confounder. If brood size is held constant, then this prediction makes

perfect sense. This problem does not arise with P_{EPO} , which is expressed as a proportion (but perhaps your PKD is as well? See previous comment.)

Author response: Here, we think that the prediction that male feeding rates should decrease with increasing PKD, now LAV, (i.e., that males should feed less when they have lost more genetic value) is an important one, and indeed an important part of our work. Please note that brood size is also explicitly accounted for in the models, and we have checked that this is clearly stated in the manuscript (Lines 254, 267).

We apologize for the confusion caused from our scaling error with PKD (LAV, see above), which may have led to your suggestion. However, to further check, we ran additional models that considered LAV/brood size. These models showed similar patterns to the main models that consider LAV and did not influence our key conclusions. We consequently did not add this information to the manuscript or supplement as we believe we have resolved the confusion with the values of LAV, and the fact that our main models do include brood size.

3. Analysis of small AIC values: The manuscript relies heavily on the Akaike Information Criterion to compare models. The authors use a cut-off of $\Delta AIC_c \geq 2$ to determine whether two models differ substantially in fit. This is already a fairly small (though commonly used) cut-off, but the authors often sneak in interpretations of Δ -values that are even smaller than this, where I would think it wiser not to interpret at all. Some examples are on lines 269-171, 310-311, 344-348.

Author response: Our original wording was used in cases where models with additional focal parameters were very weakly better supported than the competing model (i.e. ΔAIC_c between 0 and 2 units smaller). We think it is important to carefully interpret these situations, since if an additional parameter explained no variation in feeding rate then ΔAIC_c would be 2 units larger. We therefore believe that our original interpretations are reasonable. However, we have made minor edits to some instances to note that such models were similarly supported and ensure that our interpretations are not too strong (Lines 342, 356, 358, 446, 459).

4. Terminology and notation: The terms TAV, PAV, and PKD are all in the same currency of expected gene copies (I think, see comment 1). Maybe it would make more sense to rename PKD 'lost allelic value' (LAV) to make this connection clearer? Also perhaps 'total allelic value' could be renamed 'realised allelic value' (RAV) or similar. These are just suggestions.

I found the term 'total relatedness' (e.g. lines 83, 331) less than helpful, as it suggests the probability of common descent rather than the number of gene copies. You use it as a synonym for 'total allelic value', which I think is a better term.

Author response: Thank you for the suggestions about terminology. We have now changed PKD to LAV throughout. We would prefer to retain the term TAV in order to be consistent with our previous paper that presented the underlying conceptual basis for the metrics used in our current manuscript (Reid et al. 2016, Evolution). Further, we would prefer not to use the term 'realised' for this, because the quantities we calculate are pedigree expectations, not actual realisations given Mendelian sampling variance.

Based on your comments, and those of the Assistant Editor, we have now clarified that by relatedness we mean TAV. We have now added this clarification at first mention (Lines 74–76).

Responses to minor comments:

Lines 83, 87: The period for multiplication is not very standard. Please change it to a vertically centred dot (at proofs stage if not earlier).

Author response: Added vertically centred dot (Lines: 85, 89).

Line 92: Perhaps 'multiple' could be deleted?

Author response: The word 'multiple' is important and necessary in this context because a lot of the points we make require a focal breeding individual to have > 1 relative in the population (e.g. its mate and its mate's EP mate). For this reason we did not change this sentence.

Line 95-107: To me it would be more intuitive to first explain relatedness to WPO, then relatedness to EPO.

Author response: Although it may seem more intuitive to start explaining relatedness to WPO first, in the context of how we have framed this paragraph by first relating to systems where there is variation in adult-offspring relatedness it more naturally flows to start with relatedness to EPO rather than WPO. We did not make any changes to this paragraph.

Line 116: suggest changing 'as if' -> 'if'

Author response: Changed as suggested (Line: 123).

Lines 144, 207: It would help to explain why you don't expect feeding rates to depend on individuals' inbreeding coefficients. Is it because these coefficients do not differ among an individual's various brood over their lifetime?

Author response: This prediction is based on one of the key models in Duthie et al. 2017. We added some text to clarify this (Line 168). Because we clarified this point on Line 168, we felt it was not necessary to reiterate on line 253.

Lines 172-178: I can't get the numbers here to add up. Some additional explanation would help. For instance, why are there 139 nests but only 82 females? Is it because the same female can have a nest in multiple years? Do females ever have more than one nest in a single year? Also, you classify nests as monogamous/primary polygynous/secondary polygynous, but the numbers of each type of nest do not add up to the total number of nests (i.e. $82 + 30 + 29$ does not equal 139). Also, why is the number of monogamous nests the same as the number of females?

Author response: Thank you for noticing the discrepancy in the numbers. The presented numbers of monogamous nests was incorrect. This has now been corrected (Line 203). We also clarified how these numbers break down across the number of males and females and pairs (Lines 205–207). Additionally, we previously mentioned that “pairs typically rear 2–3 broods” in the paragraph before (Lines 181–182).

Lines 194, 209: Saying that an LMM 'includes' a regression on a particular parameter is a bit clumsy. The LMM is a regression. It would be nicer to say that the parameter 'was included as a covariate'.

Author response: We changed the wording of this specific sentence to reflect the comment below. Therefore, we did not change to the phrasing exactly as suggested but we did remove the word “regression” throughout and changed variable to covariate where appropriate (Lines: 241, 245, 252, 254, 267).

Line 195: What happens if you leave the effect of the social partner's feeding rate out of these models? One could argue that this you are partially controlling for a consequence of the dependent variable (although the reality is probably more nuanced).

Author response: This is a good point and something we considered when developing our models. The reason we decided to include the social partner's feeding rate was because this is one of the main factors that can influence how much a partner feeds their young as indicated by numerous experimental, empirical, and theoretical studies examining how parents negotiate parental care (e.g. Wright and Cuthill 1990, *Behav Ecol*; McNamara et al. 1999 *Nature*; Johnstone and Hinde 2006 *Behav Ecol*; Sanz et al. 2000, *J Animal Ecol*). This is now indicated on Lines 237–240.

However, in practice, when social partner feeding rate is removed from our models the key conclusions remain unchanged. We chose to leave social partner's feeding rate in our models for consistency with previous studies. We have now noted this more clearly, and also noted that key results remain quantitatively similar when this variable is removed (Lines 241–242).

Lines 222-225: It's not clear to me where these reference values come from.

Author response: This information was originally provided in the ESM Appendix 2 under “Predicted magnitude of effects”. However, after careful thought we decided that the predictions and Figure 2 were redundant as key estimated effect sizes are shown in the others figures, and Figure 2 did not provide further enhancement (and indeed was potentially confusing). We have now removed the methodology relating to the prediction methods from the text and supplement and also removed figure 2 from the main manuscript.

Line 243: The phrase 'across sessions' is a bit confusing, as at first I thought you were talking about autocorrelations of a single individual's feeding rate across different sessions. I think you are talking about the correlation between male and female feeding rate at a given nest. Maybe just delete these two words?

Author response: Deleted as suggested (Line 306).

Line 255: suggest changing 'completely' -> 'perfectly'

Author response: Changed as suggested (Line 317).

Lines 250-264: I couldn't get the delta AICc values for females to add up. Write A0, A1 and A2 for the absolute AICc values for the null model, the model with BS only, and the model with TAV only. Then I think you are saying that:

$$A1 - A0 = -12.1$$

$$A2 - A0 = -14.9$$

$$A2 - A1 = -5.0$$

But these equations are inconsistent (e.g. subtracting the first equation from the second gives $A2 - A1 = -2.8$). What's going on here?

Author response: Thank you for noticing the discrepancy. We double checked the delta AICc values in Table S2 and corrected two minor errors in the text. First, the ΔAIC_c for the female BS should have been -11.1 not 12.1 (now corrected, Line 313). Second, the ΔAIC_c value of 5.0 should have been -3.8 based on the AICc values from Table S2 (i.e., 1675.41-1679.24) Line 324.

Lines 265-268: I don't understand the need for the model without BS. Since BS is informative and presumably not highly correlated with TAVz, why not always include it?

Author response: We agree that the models with TAVz but without BS are not of primary importance. However, they do allow a further means of demonstrating the effect of BS, and allow further comparisons to the baseline model overall. For this reason we have left the models without BS in the manuscript alongside the models with BS.

Lines 285-6: The differences between these values don't look as though they would be significant. Are they?

Author response: The differences are significant based on confidence intervals and this information was presented in Appendix S5. To help clarify this we moved the statistics from Appendix S5 into the main text (Lines 351–355).

Lines 327-9: But for males the model with TAVz and BS fit slightly worse than the model with BS alone (line 268), seemingly contradicting this claim? I think your results broadly support the underlying prediction, but maybe this sentence needs to be re-worded.

Author response: The model with TAV fit better than any other model for males and females (Supplement Table S2), and this was true even after “controlling for brood size”, which refers to our model of TAV_z+BS. Our sentence is correct based on our results. To help clarify the second part of the sentence, where it may be causing the confusion, we added TAV_z in brackets (Line 526).

Lines 366-8: delete repeated phrase

Author response: Repeated phrase deleted as suggested (Line 576).

Fig 2 caption: Please include reference to the lower bound of zero for PKD.

Author response: We removed Figure 2 (please see response above).

Referee: 2

Comments to the Author(s)

This is a well conceived and well executed study relating offspring-parent genetic relatedness (which can be different for males and females due to extra-pair paternity), within-breeding pair relatedness and parental inbreeding coefficient to parental provisioning (i.e. feeding rate) by using a well established biparental bird system. This study adds a very interesting facet to the already impressive work of the group. The design is clear and well thought and the data seem to be analyzed appropriately. The MS is very well written. As predicted by kin selection theory, feeding rate of both sexes increases with increasing relatedness to the offspring but increasing within-pair relatedness increases feeding rates of males only. The results are very interesting and should appeal a broad readership.

Author response: Thank you very much for the positive review and praise.

Comments

*-What could be the underlying kin recognition mechanism allowing such fine-tuned differentiations (especially when taken into account multiple paternities): maternal imprinting, self-reference? Does offspring behavior play a role, e.g. half sibling broods might differ from full-sibling broods and that could be used as proxy for paternal relatedness.
-Does variation in quality (e.g. body size) within and among pairs play a role? Quality assortative mating might increase within-pair genetic relatedness.*

Author response: These are good questions, and for song sparrows we do in fact have some knowledge about how individuals could evaluate their relatedness to each other, and hence to a dependent brood. Specifically, previous studies in song sparrows have shown that relatedness can be indicated by song repertoire size (Reid et al. 2005 PRSB), population demographic structure (Reid et al. 2015 JAE), and preen gland oil (MHC) composition (Slade et al. 2016 PRSB). In our original manuscript we had originally noted these mechanisms of kin recognition and studies at the end of the discussion, and we have checked this is clear in the revised version (Lines 626–627). Further, given some of your suggestions, we also added a clause at the end stating “but of course other mechanisms, such as differential offspring behaviour, might also be involved”, (Lines 627–628).

-How reliable is the feeding rate? Is it a reliable predictor of the amount of food provided or is it prone to cheating?

Author response: From our field observations feeding rate appears to be a generally reliable predictor of the amount of food provided. This is because parents were not observed to visit the nest without food, and most visits consisted of 1–4 items, which varied in size (e.g. when there were fewer items they were usually large and when more items they were usually small). To clarify these points we added some sentences in the ESM Appendix S2, “parental feeding rates” section (Lines 136–139).

-This study is not the first one that shows the interrelation between inbreeding, genetic relatedness of parents and parental brood care. In 2007 Thunken et al. published the experimental study "Active inbreeding in a cichlid fish and its adaptive significance" (Current Biology) in which the authors showed kin mating preferences in a biparental cichlid fish and proposed a potential benefit of inbreeding that had not been addressed so far: that the higher genetic relatedness between parents might improve parental care because of a reduced sexual conflict over care due to kin selection. In accordance with this idea the study also showed that related parents, i.e. full siblings, spent more time protecting the young than unrelated parents and that males were less aggressive when the female partner was a relative. Although the paper is mentioned as reference for active inbreeding (lines 93 and 376), its pioneering aspect (link between inbreeding, kin selection and parental care) relevant for the present study should be acknowledged.

Author response: Thank you for the suggestion about further emphasizing the Thunken et al. study. We have now further referenced this study in the discussion (Lines 582–583).

-the high number of abbreviations makes it a bit difficult to read the ms, especially the discussion. Maybe add a table summarizing the abbreviation.

Author response: We understand the concern of too many abbreviations. We have done our best to only use a minimum number and have included all the abbreviations in Table 1, to provide an easy reference. Once the manuscript is typeset this table should appear early (rather than at the end of the submitted document) and should then become easier to refer to.

Furthermore, we have also defined key variables again at their first mention in the Discussion (TAV: Line 525; LAV: Line 535; k : Line 535) and have now added a definition for P_{EPO} (Line 534). We have also now added the abbreviations to the subheadings in the results (Lines: 311, 346, 441). Further, given the suggestion of Reviewer 1 to change PKD to LAV (lost allelic value) we think this will also help the reader follow the terminology.